# Injectable Capsaicin for the Management of Pain Due to Osteoarthritis

**DOI:** 10.3390/molecules26040778

**Published:** 2021-02-03

**Authors:** James N. Campbell, Randall Stevens, Peter Hanson, James Connolly, Diana S. Meske, Man-Kyo Chung, Benedict Duncan X. Lascelles

**Affiliations:** 1Centrexion Therapeutics, Boston, MA 02109, USA; jcampbell@centrexion.com (J.N.C.); rstevens@centrexion.com (R.S.); jconnolly@centrexion.com (J.C.); dmeske@centrexion.com (D.S.M.); 2eGenesis, Cambridge, MA 02139, USA; peter.hanson@egenesisbio.com; 3Center to Advance Chronic Pain Research, Program in Neuroscience, The Department of Neural and Pain Sciences, School of Dentistry, University of Maryland, Baltimore, MD 21201, USA; MChung@umaryland.edu; 4Translational Research in Pain (TRiP) Program, Comparative Pain Research and Education Centre, Department of Clinical Sciences, College of Veterinary Medicine, North Carolina State University, Raleigh, NC 27606, USA; 5Thurston Arthritis Centre, UNC School of Medicine, Chapel Hill, NC 27599, USA; 6Center for Translational Pain Research, Department of Anesthesiology, Duke University, Durham, NC 27710, USA

**Keywords:** intra-articular, cooling, knee, defunctionalization, nociceptive fiber, TRPV1, primary afferent, nociceptor, disruption

## Abstract

Capsaicin is a potent agonist of the TRPV1 channel, a transduction channel that is highly expressed in nociceptive fibers (pain fibers) throughout the peripheral nervous system. Given the importance of TRPV1 as one of several transduction channels in nociceptive fibers, much research has been focused on the potential therapeutic benefits of using TRPV1 antagonists for the management of pain. However, an antagonist has two limitations. First, an antagonist in principle generally only affects one receptor. Secondly, most antagonists must have an ongoing presence on the receptor to have an effect. Capsaicin overcomes both liabilities by disrupting peripheral terminals of nociceptive fibers that express TRPV1, and thereby affects all of the potential means of activating that pain fiber (not just TRPV1 function). This disruptive effect is dependent on the dose and can occur within minutes. Thus, unlike a typical receptor antagonist, continued bioavailability at the level of the receptor is not necessary. By disrupting the entire terminal of the TRPV1-expressing nociceptive fiber, capsaicin blocks all the activation mechanisms within that fiber, and not just TRPV1 function. Topical capsaicin, an FDA approved treatment for neuropathic pain, addresses pain from abnormal nociceptor activity in the superficial layers of the skin. Effects after a single administration are evident over a period of weeks to months, but in time are fully reversible. This review focuses on the rationale for using capsaicin by injection for painful conditions such as osteoarthritis (OA) and provides an update on studies completed to date.

## 1. Introduction

The idea that capsaicin, the pungent alkaloid of hot peppers (Capsicum annuum), might be used to treat pain at first seems counterintuitive. How can a molecule that causes pain also treat pain? Though capsaicin indeed induces a temporary painful sensation, the more enduring effect at sufficient doses is pain reduction. Capsaicin in an 8% concentration (Qutenza) is approved by the US Food and Drug Administration (FDA) for the management of postherpetic neuralgia and by the European Medicines Agency (EMA), more broadly, as a topical therapy to manage peripheral neuropathic pain [1,2,3]. Topical therapy addresses the pain that stems from abnormal nociceptive fiber (pain fiber) activity at the level of the skin. Despite the proven benefits of capsaicin as a topical treatment, there is still no approved use of capsaicin as an injection therapy to manage pain that arises from deeper structures. In this review we will provide an update on the use of injectable intra-articular capsaicin to manage painful osteoarthritis (OA).

Capsaicin induces pain via the transduction channel, transient receptor potential cation channel subfamily V member 1 (TRPV1). Several excellent reviews discuss the biology of TRPV1 and interactions with ligands such as capsaicin [4,5,6,7]. For present purposes, it is to be noted that TRPV1 is an ion channel that is expressed selectively on a substantial proportion of nociceptive fibers that signal pain. Many studies indicate that activity in nociceptors (or the peripheral terminals of nociceptive fibers) is associated with the sensation of pain, and that nociceptor sensitization is an important mechanism in the production of pathological pain [8]. Though widely expressed, not all nociceptive fibers express TRPV1 [9,10]. The identification and cloning of TRPV1 was a major landmark in stimulating a new area of research in the field of pain [11]. The knockout of TRPV1 expression, through genetic tools, eliminates the pungency of capsaicin, thus confirming that TRPV1 solely mediates the sensory effects of capsaicin [12]. TRPV1 has a central ion-conducting pore that is formed by four homomeric subunits; each subunit is formed by six transmembrane helices [13]. Capsaicin is a highly lipophilic molecule that passes rapidly from the extracellular space to the intracellular region where it binds to a transmembrane portion of TRPV1. Binding leads to a conformational change resulting in opening of the pore [14]. When the TRPV1 pore is opened by capsaicin or other ligands, calcium and sodium ions flow into the cell as governed by their concentration gradients. This influx of positive cations (an inward current) induces depolarization, which in turn leads to activation of voltage gated sodium channels, followed by initiation of action potentials. The action potentials generated by TRPV1 mediates nociception (“the neural process of encoding noxious stimuli”) in peripheral terminals of nociceptive fibers. Activation of nociceptors can lead to pain (“an unpleasant sensory and emotional experience associated with, or resembling that associated with, actual or potential tissue damage”) depending on context, the level of activation, and other complex central neural mechanisms [5,15].

## 2. Rationale for the Therapeutic Use of Capsaicin for Pain Management

In conducting experiments with capsaicin in human subjects, it became apparent that one could obtain complete analgesia to heat in the area of an intradermal injection of capsinoid molecules [16]. This effect was specific to the area of injection. If one studied the area millimeters away from the injection site, there was no effect [17]. The question emerged as to whether this could be taken advantage of therapeutically. This work led to the development of high dose topical capsaicin, as noted above.

Simone and colleagues performed studies in human subjects wherein intradermal doses of capsaicin ranging up to 20 µg were administered. Punch biopsies from the sites of injection and the immediately adjacent area were obtained and stained with the pan-axonal marker, protein gene product 9.5 (PGP 9.5), to determine the presence of epidermal C fibers, the majority of which are likely to be nociceptive afferents (Figure 1). These TRPV1 epidermal nerve fibers were disrupted (referred to in other publications as “ablation”) as a function of dose, with total disruption or ablation evident with an injection of 20 µg. Further punch biopsies were obtained over the following weeks. These studies indicated recovery of the TRPV1 nerve fibers over the ensuing weeks. This disruptive effect was highly restricted spatially. Within millimeters of the injection, there was no effect on nerve fiber counts [17]. This is consistent with the highly lipophilic properties of capsaicin. Capsaicin is unlikely to spread from the epidermis to deeper layers beyond the skin, and is likely taken up into the circulation due to the high degree of lipophilicity. Of note is that the stratum corneum is rich with lipids [18]. Psychophysical studies indicate an effect on sensation consistent with the findings of these immuno-histological studies whereby heat-pain sensibility is eliminated in the area of the injection followed by sensory recovery over time.

Capsaicin-induced disruption of nerve terminals is also evident in in vitro preparations. Appendix A represents experiments using a microfluidic chamber with sensory neurites that have been genetically engineered to express membrane-bound green fluorescent protein (GFP) under the promoter of TRPV1 [19]. Capsaicin (100 µM) was applied, and one can see the local response of the putative nociceptive afferent neurites. In response to capsaicin, neurites beaded and were then disrupted. Since capsaicin was applied to the axonal chamber, these effects were restricted to the TRPV1-expressing nerve fibers without affecting soma. Non-TRPV1-expressing neurites were unaffected by capsaicin (Appendix A).

These experiments demonstrate that the effects of capsaicin occur within minutes of application and confirm the specificity of the effects in terms of the types of nerve fibers that are affected. The precise mechanism by which capsaicin disrupts TRPV1-expressing nociceptive terminals remains unclear. Influx of extracellular Ca^2+^ through the TRPV1 pore is required. Ca^2+^ influx into the nociceptive terminal activates calpains, and Ca^2+^-dependent proteases, which is ultimately associated with disruption of the axonal cytoskeletal proteins [19]. However, the disruptive effect is likely to entail other mechanisms as well. Other means of activating TRPV1 (e.g., heat stimuli) do not have the same disruptive effect.

While this could merely be a function of the degree of activation, TRPA1 agonists, such as mustard oil (allyl isothiocyanate), also mediate an influx of calcium. However, TRPA1 agonists do not appear to cause disruption even when the pain associated with application is matched to that induced by capsaicin application given at a dose that does induce disruption [20]. Capsaicin mediated disruption at the terminals of nociceptive fibers, therefore, probably involves more than just calcium influx. Leading candidates include mitochondrial mechanisms, and direct effects on microtubules linked specifically with TRPV1 [5]. Despite an incomplete understanding of the mechanisms, a recent preclinical study verified that disruption of TRPV1-expressing afferent terminals by capsaicin is necessary for long-lasting analgesia related to neuropathic pain [21]. The conclusion is that disruption of TRPV1-expressing terminals of nociceptive afferents likely underlies the long-term analgesia produced by capsaicin.

The initial discovery of the TRPV1 transduction channel, and its role in pain, led to a major push to identify specific antagonists of TRPV1. The hope was that an oral antagonist would be an effective analgesic. Side effect and efficacy issues, however, have thwarted efforts to date [22]. TRPV1 is only one of many ion channels expressed on TRPV1-expressing pain fibers [23]. The difficulties encountered in developing novel therapeutics could reflect, in part, this multiplicity of mechanisms. One antagonist can block one channel, but then another can compensate. Capsaicin, however, does not block the channel. Capsaicin defunctionalizes the nerve fiber terminal. With disruption or ablation of the nerve fiber endings, all means of activating the terminals of nociceptive fibers are eliminated. In a sense, this is a Trojan horse mechanism. The capsaicin gets “in the door” and then deactivates the pain fiber. These effects are fully reversible in the ensuing months, as the nociceptor regenerates to normal levels over time [17].

One can appreciate this phenomenon from a common culinary experience. Pungency is experienced with eating hot peppers. However, with repeated exposure over weeks and months, people develop a tolerance and can eat hot peppers in much greater amounts. [24] However, after a “hot pepper” holiday the sensitivity to hot peppers returns. These observations fit with the concept that capsaicin induces a reversible disruptive effect on the nociceptors expressed in the oral cavity.

It is noteworthy that a therapeutic dose of capsaicin or the amount of capsaicin in foods is not high enough to produce permanent neuronal death in sensory ganglia. Although preclinical studies suggest that high dose of systemic capsaicin can induce degeneration of sensory neurons within sensory ganglia, cutaneous or intrathecal injection of capsaicin produces only localized disruption (ablation) of axonal terminals without neuronal death in ganglia. [19,25] The doses of capsaicin administered for treating pain in humans is even lower by orders of magnitude. Therefore, the aforementioned disruption and regeneration of axonal terminals following capsaicin injection occurs in the site of capsaicin injection rather than involving the entire length of peripheral axons or sensory ganglia. The possibility exists that the nerve fibers that reinnervate the targeted sites of capsaicin delivery in the periphery may also be derived from ramification of unaffected axonal terminals adjacent to the site of capsaicin injection. It is also an open question if the regenerated fibers include non-TRPV1-expressing fibers. However, the effectiveness of repeat delivery of topical capsaicin to treat neuropathic pain and the return of capsaicin sensitivity in the mouth after a period of a capsaicin free diet suggest that the reinnervating fibers include fibers expressing TRPV1 [26].

An important point to highlight is that many nociceptors do not express TRPV1. [27] The implications of this are two-fold: capsaicin may not be effective across all types of pain, and conversely, capsaicin will not cause complete defunctionalization.

## 3. Current Therapies for Osteoarthritis

OA is often accompanied by pain. Indeed, one learns from talking to patients that the chief complaint associated with OA is pain. In the absence of pain, the underlying OA may not even be evident to the patient. It follows that a treatment that addresses the pain meets the needs of patients even if it does not improve the underlying OA.

Current therapies for OA pain face major limitations. Definitive treatment typically requires a total joint replacement—not a trivial undertaking. Total knee joint replacement is one of the most common operations performed, and the prevalence in the USA in individuals over the age of 50 has been estimated to be 4.6% [28]. The outcome is not uniformly favorable. About two-thirds of patients still have pain with knee movement 18 months after surgery [29]. A substantial percentage of patients are not satisfied with the results [30]. Only 22% of patients rate their results as excellent, and only 71% report being “much improved” [31]. On top of this, undergoing a major operation, such as a knee replacement, is not an option for many patients due to complicating medical conditions and socioeconomic issues. Oral pharmacologic therapies also have limitations. Nonsteroidal anti-inflammatory drugs (NSAIDs) likely represent the most efficacious oral pharmacologic therapy. However, over time these effects wane [32]. The peak effects are evident at two weeks and clinical significance may be lost as early as 8 weeks after initiating the therapy. In addition, there are safety risks that include gastrointestinal, renal, and cardiovascular morbidities and mortalities. Opioids also have clear analgesic efficacy; however, particularly when used for chronic pain, they are not without serious safety risks (e.g., abuse liability, respiratory depression/overdose) and tolerability issues (e.g., constipation, tachyphylaxis) [33]. Injections of corticosteroids and hyaluronic acid offer short-term benefits, but do not improve symptoms when compared to placebo in the long run and may even be associated with worsening of symptoms [34]. Longer term use of corticosteroids has been associated with cartilage volume loss [35].

## 4. Capsaicin Injection for Osteoarthritis

TRPV1 has been implicated in knee joint nociceptor sensitization, hyperalgesia, or knee joint edema from preclinical OA models [36,37,38]. To examine the effects of capsaicin administered intra-articularly (IA) into the knee joint on OA pain, a dose ranging, placebo-controlled, randomized, double-blind Phase 2 study was conducted [39]. In this study, the analgesic effects and safety of placebo, 0.5, and 1.0 mg of capsaicin (CNTX-4975, specifically) were compared [39]. Subjects were followed for 6 months (24 weeks) following the IA injection. The primary endpoint was based on a patient’s rating of “pain with walking on a flat surface” (Western Ontario and McMaster Universities Osteoarthritis Index, question A1). This was recorded in a daily diary using an 11-point numerical pain rating scale (NPRS) [40]. The primary endpoint was analyzed using an area under the curve (AUC) analysis based on the outcome at 12 weeks (note: 12 weeks is the standard surrogate endpoint for chronic analgesic therapies used for approval by the FDA and EMA). Radiograph severity of knee OA structure was evaluated using the Kellgren and Lawrence (KL) scale (0–4). [41] Grades 2 to 4 were accepted into the study and a body mass index of up to 45 Kg/m^2^ was allowed. The alpha was set at 0.1 a priori in this study due to the small sample size. At week 12, greater decreases in the AUC for pain with walking was observed with capsaicin in the 0.5 and 1.0 mg groups versus placebo (0.5 mg group least squares mean difference (LSMD): −0.79, P = 0.0740; 1.0 mg group LSMD −1.6, P < 0.0001). Statistically significant improvements were maintained at week 24 in the 1.0 mg group (LSMD: 1.4, P = 0.0002). In Figure 2, the average daily pain with walking on a flat surface over time is plotted based using a mixed model for repeated measures analysis (MMRM). The mean pain level at the beginning of the study averaged from 7.2 to 7.4 on an 11-point NPRS for the three groups. Consistent with the AUC analysis, the decrease in pain at 12 weeks was significantly greater in the 1 mg group compared to the placebo group (P < 0.001). The effects of the 0.5 mg dose were intermediate between the placebo and 1 mg dose (i.e., indicative of a dose response). The 1 mg treatment demonstrated statistically significant improvement over placebo at almost all time points out to 24 weeks.

The WOMAC scale includes assessments of stiffness (WOMAC B subscale), and function (WOMAC C subscale) (the WOMAC A scale was not administered in this study). These endpoints are commonly used as endpoints in knee OA clinical trials. Both the stiffness (WOMAC B) and function (WOMAC C) scales demonstrated statistically significant improvement for the 1mg treatment group, as compared to placebo, and that effect was maintained out to week 16 [39]. In addition, numerical superiority was evident out to week 24. Responder rates were also assessed in secondary analyses. Responder rates provide meaningful insight into the analgesic efficacy of a product, as there are commonly accepted cutoffs that apply clinically meaningful changes (≥30% and ≥50% improvement) [42]. Over 60% of subjects in the 1mg treatment group had a ≥50% reduction in pain versus only 33% of subjects in the placebo group, equating to a number needed to treat (NNT) value of 3.6 at 12 weeks. In the 1 mg group, 20% of subjects had a ≥90% reduction in NPRS pain scores. There were no identified safety issues during the conduct of this study, as demonstrated by a similar incidence of adverse events in all treatment groups.

## 5. Injection Versus Topical Delivery

The question might be asked as to whether it is necessary to inject directly into the joint to achieve efficacy for OA pain. Would it be possible to simply place capsaicin topically over the knee and achieve similar efficacy? In the study of Stevens et al. discussed above [39], the 0.5 mg dose was borderline in terms of statistical significance for the primary endpoint (pain on walking at 12 weeks). Therefore, it is likely that 1 mg directly in the joint is required to achieve clinical benefit. To estimate joint delivery from topical application, it is worthwhile to consider the NSAID diclofenac. Diclofenac is approved for topical delivery as a means of treating painful OA of the knee [43]. Seefried and colleagues [44] studied diclofenac (2.23% *w*/*w* gel) penetration into the knee joint after topical delivery at doses known to be effective in managing knee OA pain (74.4 mg applied twice daily for 7 days). The mean concentration of diclofenac was 1.57 ng/g in the synovial tissue, and 2.27 ng/mL in the synovial fluid. This represents a topical application to synovial fluid concentration ratio in the order of one million. If one assumes that 1 mg dosing administered to the joint leads to a synovial fluid concentration of approximately 200 µg/mL (based on 5 mL of synovial fluid; 655 µM), a greater than 200,000 mg dose of capsaicin would need to be applied to the skin. Topical dosing with 1 mL at a concentration of 10% capsaicin would supply 100 mg of capsaicin, clearly orders of magnitude less than what it would take to provide a concentration of 200 µg/mL.

These calculations are based on diclofenac, an amphiphilic molecule that has favorable hydrophilicity consistent with deeper delivery. To be absorbed into deeper tissues a drug ideally needs to have some level of water solubility [45]. Capsaicin has unfavorable physiochemical properties for deep delivery into the tissues given that it is strongly lipophilic/hydrophobic [46]. The effects of injection of capsaicin directly into the skin are confined to the site of injection as described above (Figure 1). Photomicrographs from human skin biopsies analyzed through quantitative counts of the epidermal nerve fiber density indicate that effects are restricted precisely to the site of the injection [17]. At 1 or more millimeters from the injection site, the density of the epidermal fibers was normal, arguing against potential diffusion of capsaicin to adjacent tissues. After initial binding in the skin, the capsaicin most likely is simply taken up in the general circulation. Given that 1 mg injected into the joint is required for efficacy, it seems implausible that one would derive meaningful benefit from a topical dose for the treatment of pain associated with knee OA. Although metabolism of capsaicin in the skin, per se, is slow [47], there is constant removal from the skin into the systemic circulation, limiting local build up.

## 6. Cooling for Decreasing Procedural Pain

One of the challenges in using capsaicin is that it induces transient pain [48]. When capsaicin is injected intradermally, the discomfort associated with injection peaks within 1 min, and at the highest doses decreases to a minimal level by 15 min [49]. With IA injection of capsaicin, the pain lasts longer and is still evident at 30 min. This observation of discomfort on injection of capsaicin indicates that a substantial population of sensory afferents in the joint must express TRPV1. Of note, in the study of Stevens et al. [39] there was also procedural pain in the placebo group, though at a reduced average level. Of note, the full range of pain was seen in both placebo and capsaicin treatment arms. Moreover, within each group (placebo, 0.5 mg, 1 mg) the subsequent decrease in OA pain had no relation to the extent of procedural pain (i.e., procedural pain was not correlated with primary endpoint data).

To control the pain associated with IA injection of capsaicin, preinjection with lidocaine was found to be helpful, but still not optimal. It has been reported that the pain from topical delivery of capsaicin to the skin can be reduced using cooling. Importantly, the disruptive or ablative effect was preserved as demonstrated with quantitative nerve fiber counts [50]. To achieve cooling for IA injection, we initially used a device tethered to a circulating ice water bath. Subsequently, we performed a study with probes placed into the knee joint and determined that approximately the same amount of intra-articular cooling occurred with an ice gel pack wrapped around the knee [51]. The ice gel pack is obviously much easier to use and allows the person to ambulate with the cooling device in place. In this work [51], we discovered that cooling the joint was significantly correlated with a reduction in capsaicin related procedure pain (Figure 3).

## 7. Experience with Bilateral Knee Injection

Many patients experience substantial pain in both knees. In a separately performed open label safety study reported elsewhere [52], 848 subjects with painful OA of the index knee received capsaicin 1 mg via IA injection into the knee (ClinicalTrials.gov, Identifier: NCT03661996). Of these, the majority had pain in the opposite side and received a second injection into the opposite knee. This second injection in the opposite knee was performed 8 days after the first injection into the index knee (n = 427). Pain with walking was assessed either in the clinic or by telephone follow up on a 11-point NPRS. The results shown in Figure 4 indicate that the pain reduction of approximately 4 points was similar for the index knee (mean baseline pain: 7.4) and for the opposite side (mean baseline pain: 6.1). The magnitude of the pain reduction was similar to the results of the double-blind randomized controlled study reported by Stevens et al [39]. These data suggest that patients with bilateral pain can be treated on both sides within a short time frame.

## 8. Expanded Opportunities

The knee is not the only joint affected by OA. Once definitive Phase 3 studies are completed, and approval is obtained for treatment of the knee OA pain, attention can be turned to the treatment of other joints. Many joints are not amendable to joint replacement. One example is the metacarpophalangeal joint at the base of the thumb where the value of corticosteroid injections and other conservative methods of treatment have limited success [53,54]. Similar opportunities for other joints throughout the body exist, and even potentially the spine. Beyond humans, opportunities exist in veterinary medicine for providing long-lasting, safe pain relief to selected joints. Preliminary studies have suggested efficacy of IA capsaicin in dogs with naturally occurring OA.

## 9. Conclusions

The leading complication of OA from the patient perspective is pain. Capsaicin is a highly selective molecule that has a disruptive effect when administered to the terminals of TRPV1-expressing nociceptive sensory fibers—effectively causing ablation of the terminals of the sensory fibers. Over a period of months, this disruptive effect disappears, as the effects of capsaicin are reversible. The potential exists that clinicians can take advantage of the disruptive effects by delivering capsaicin directly to sites that generate pain. The high degree of selectivity, the rapid effects, and the safety record to date suggest that capsaicin injection for control of OA pain has great promise. Focus to date has been on the knee joint, given the large unmet need and the fact that injection into the joint can be carried out as a relatively simple office procedure. Procedure pain has been an obstacle to capsaicin development in the past. Present data suggest, however, that with preemptive cooling, procedure pain can be reduced substantially. For each of these reasons, capsaicin has promise as an important means to control OA pain.

## Figures and Tables

**Figure 1 molecules-26-00778-f001:**
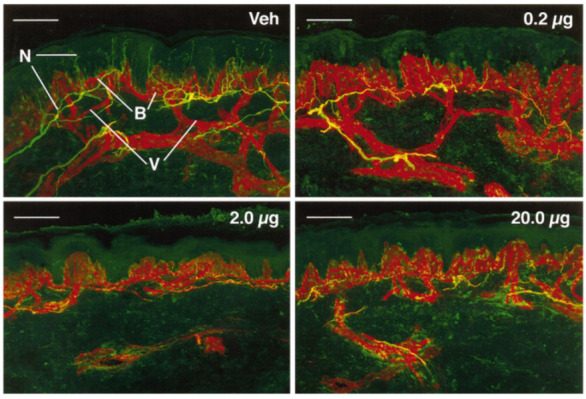
(Reprinted from [17], Copyright 1998 Society for Neuroscience) Confocal images of human epidermal biopsies from one individual who received vehicle and varying doses of capsaicin. Nerve fibers (N) immunoreactive for PGP 9.5 appear yellow-green, and basement membrane (B) and vessels (V) appear red. Biopsies were taken 72 h after injection of vehicle (Veh) or capsaicin doses of 0.2, 2.0 or 20 μg. After the lowest dose of capsaicin, loss of PGP 9.5-immunoreactive nerve fibers was restricted to fibers located in the epidermis. Higher doses of capsaicin resulted in complete loss of PGP 9.5-immunoreactive epidermal nerve fibers plus various degrees of disruption or complete loss of the subepidermal nerve plexus. Scale bars = 100 μm.

**Figure 2 molecules-26-00778-f002:**
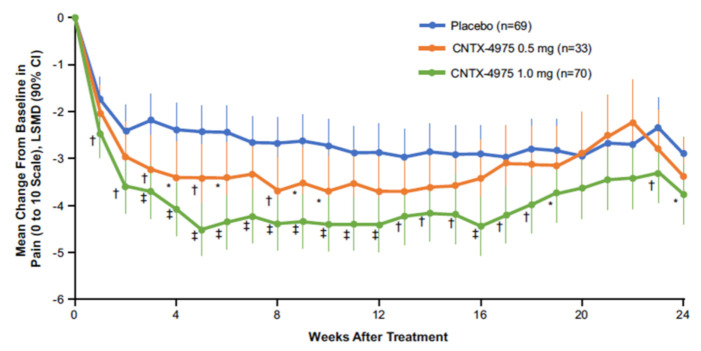
(Reprinted with permission from [39]) Plot of the change from baseline in average weekly WOMAC pain with walking scores through week 24 in patients treated with intra-articular capsaicin (CNTX-4975, specifically) versus placebo is shown. A mixed model for repeated measures was used in the modified intent-to-treat population. Week 12 was the prespecified landmark endpoint. Baseline scores (range 0–10): placebo 7.4, CNTX-4975 0.5 mg 7.2, CNTX-4975 1.0 mg 7.2. * = P < 0.1; † = P < 0.05; ‡ = P < 0.001 versus placebo.

**Figure 3 molecules-26-00778-f003:**
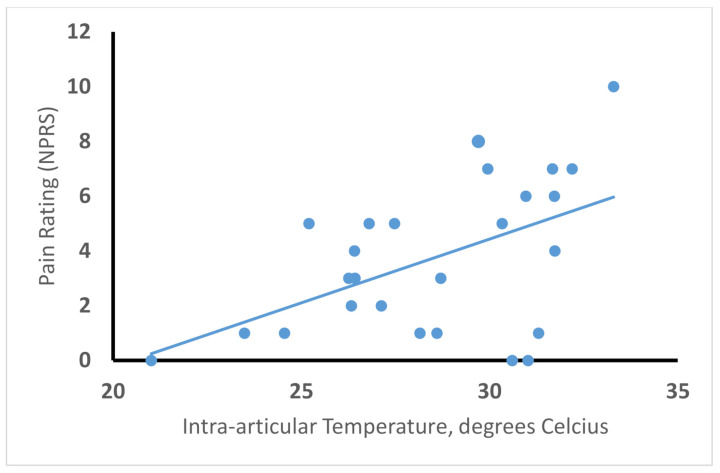
Procedural pain was significantly and positively, associated with intra-articular (IA) temperature. In each of 13 individuals with knee OA, different techniques for cooling the left and right knee joints were used, resulting in varying IA temperatures (measured by an IA temperature probe). IA lidocaine, followed by capsaicin (1mg), was administered and procedural pain (on a Numerical Pain Rating Scale, NPRS) and IA temperature were measured at intervals following the injections. The plot of NPRS versus IA temperature demonstrated a positive (r = 0.51) and significant (P = 0.008) correlation. These data support the use of joint cooling as a means to decrease procedural pain associated with IA capsaicin injection. Number of subjects = 13; both knees represented for each subject.

**Figure 4 molecules-26-00778-f004:**
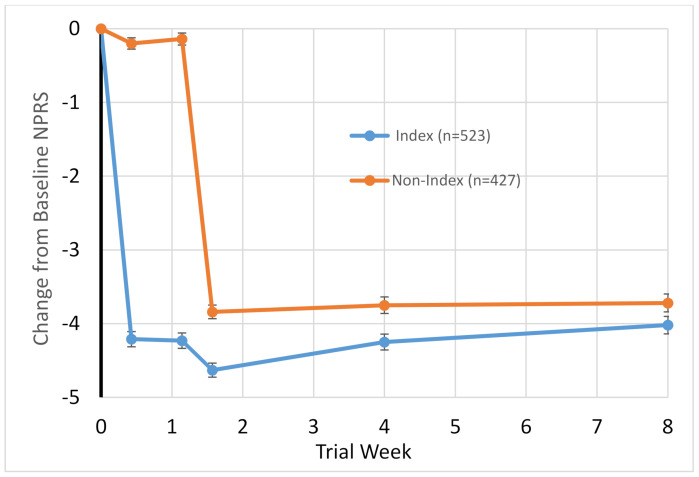
Graph of the change in pain while walking (assessed using an 11-point NPRS on a scale of 0–10) in patients with bilateral knee OA undergoing treatment of the index knee (n = 523) and treatment of the nonindex knee (n = 427). This second injection in the opposite knee was performed 8 days after the first injection into the index knee. The magnitude in pain reduction was similar for the index and nonindex knee.

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
