# Peer review of "Injectable Capsaicin for the Management of Pain Due to Osteoarthritis"

_molecules, 2021, doi:10.3390/molecules26040778_

Round 1

Reviewer 1 Report

The manuscript entitled “Injectable Capsaicin for the Management of Pain due to osteoarthritis”, submitted by Campbell et al, have reviewed the current update of injectable intra-articular capsaicin for the management of pain in patients with osteoarthritis. They have clearly described the mechanisms of how capsaicin induces pain while intradermally injected and potential involvement of TRPV1 channel in nociceptor of skin. The application of capsaicin after intra-articular (IA) injection can also result in pain relief in a dose dependent manner in patients with osteoarthritis. However, in current manuscript, the connection and the rationale of applying capsaicin in the treatment of osteoarthritis-induced pain is likely not sufficient. The authors should provide more substantial evidence/rationale of capsaicin in pain relief of osteoarthritic pain. In addition, the reason of causing pain after IA of capsaicin in relieving osteoarthritic pain is not clear. It is better if the authors can expand their speculation on how this capsaicin-resulted elongated pain.

Author Response

Reviewer 1

The manuscript entitled “Injectable Capsaicin for the Management of Pain due to osteoarthritis”, submitted by Campbell et al, have reviewed the current update of injectable intra-articular capsaicin for the management of pain in patients with osteoarthritis. They have clearly described the mechanisms of how capsaicin induces pain while intradermally injected and potential involvement of TRPV1 channel in nociceptor of skin. The application of capsaicin after intra-articular (IA) injection can also result in pain relief in a dose dependent manner in patients with osteoarthritis. However, in current manuscript, the connection and the rationale of applying capsaicin in the treatment of osteoarthritis-induced pain is likely not sufficient. The authors should provide more substantial evidence/rationale of capsaicin in pain relief of osteoarthritic pain. In addition, the reason of causing pain after IA of capsaicin in relieving osteoarthritic pain is not clear. It is better if the authors can expand their speculation on how this capsaicin-resulted elongated pain.

REPLY: Thank you very much for the review and the helpful comments. We have added language in the abstract and other locations that makes the rationale for using capsaicin more clear. In addition, we added information about preclinical studies that support the use of capsaicin for knee osteoarthritis (section 4). We also have clarified that the therapeutic effects are dependent on the disruption of the function of nociceptors, as reflected in the video clips presented in this manuscript, and prior immunohistological studies (republished here).  The reviewer correctly highlights the relatively small amount of published data presently available regarding the efficacy of capsaicin in osteoarthritis.  However, the data presented has been published in a top tier peer reviewed journal (Stevens et al, 2019).  Moreover, the authors point to the ongoing human trials aimed at further corroborating the promise of this emerging novel therapy.  The relief of osteoarthritic pain is not dependent on causing pain. We have added text to Section 6 that further clarifies that the induction of pain by capsaicin, and the therapeutic effect are independent. 

Reviewer 2 Report

The narrative review by Campbell and colleagues is an excellent piece of work that will be useful for those working clinically with capsaicin.

I have only a few minor comments:

  1. In the abstract (line 2) and other parts of the paper the authors indicate the nociceptors as the pain fibers. Although such a definition is widely used clinically, it is somewhat misleading, as strictly speaking, nociceptors are either the neurons giving rise to these fibers (i.e., the neuronal cell bodies in DRGs in the case of joints) or the terminal differentiations of these fibers that should be correctly indicated as nociceptive fibers.
  2. There is indeed a difference between nociception and pain (as reviewed e.g., in Molecules. 2016 Jun 18;21(6):797 doi: 10.3390/molecules21060797) and such a difference is not mentioned in the paper. I suggest that authors make a mention to this, maybe even simply in a footnote placed wherever it may be more convenient.
  3. On several occasions, the authors use the term ablation to indicate the effects of local capsaicin onto tissue. The term is somewhat misleading. Indeed, what capsaicin induces is a temporary afferent denervation of these tissues. Also, it should perhaps worth mentioning that the vanilloid induces a permanent afferent denervation under certain experimental conditions in preclinical studies and that there is, theoretically, the possibility that the reinnervation of tissue biopsies (figure 1 form ref. 16) derives form fibers others than those degenerated because of capsaicin treatment (see Neuroscience. 1990;39(2):501-11. doi: 10.1016/0306-4522(90)90286-d).
  4. When describing the results of IA capsaicin injection for treating osteoarthritis (page 5 line 10) the authors indicate as statistically significant a P=0.0740. This is not correct. Indeed, they correctly remark in the subsequent text that the results of 0.5 mg were simply indicative of an effect, but as such the text is confusing to readers.
  5. It should also be useful mentioning prospectively that preclinical studies have indicated the existence of a large proportion of polymodal C-fibers that do not express TRPV1 and it appears that different types of stimuli (thermal versus mechanical/chemical) can activate these TRPV1−negative non-peptidergic C-nociceptors (Cell Mol Neurobiol. 2020 Apr 18. doi: 10.1007/s10571-020-00847-w. Online ahead of print). If this may be translated clinically, it would imply that capsaicin will be ineffective in suppressing all types of pain when given in osteoarthritis or other conditions.

Author Response

Reviewer 2

The narrative review by Campbell and colleagues is an excellent piece of work that will be useful for those working clinically with capsaicin.

REPLY: Thank you very much for the review and comments.

I have only a few minor comments:

  1. In the abstract (line 2) and other parts of the paper the authors indicate the nociceptors as the pain fibers. Although such a definition is widely used clinically, it is somewhat misleading, as strictly speaking, nociceptors are either the neurons giving rise to these fibers (i.e., the neuronal cell bodies in DRGs in the case of joints) or the terminal differentiations of these fibers that should be correctly indicated as nociceptive fibers.

REPLY: We agree that an argument can be made that in many cases reference to “nociceptive fibers” may be a preferred term as opposed to nociceptor.  However, the lead author (JNC) has published many tens of studies with thousands of citations in the basic science literature (for example, Science, Journal of Neuroscience, Neuron, leading textbooks) that refer to nociceptors in the same way as proposed in this text.  Reference to a “nociceptive fiber” does not make clear that the target is the terminal where the transduction mechanisms are fully expressed.  As an acknowledgement of the reviewer’s comment, we have added text that refers to “nociceptive fibers” in certain locations in the text to complement the use of nociceptors as a term. 

  1. There is indeed a difference between nociception and pain (as reviewed e.g., in Molecules. 2016 Jun 18;21(6):797 doi: 10.3390/molecules21060797) and such a difference is not mentioned in the paper. I suggest that authors make a mention to this, maybe even simply in a footnote placed wherever it may be more convenient.

REPLY: Thank you for this comment, and we have added into the paper (first paragraph) a sentence to point this out

  1. On several occasions, the authors use the term ablation to indicate the effects of local capsaicin onto tissue. The term is somewhat misleading. Indeed, what capsaicin induces is a temporary afferent denervation of these tissues. Also, it should perhaps worth mentioning that the vanilloid induces a permanent afferent denervation under certain experimental conditions in preclinical studies and that there is, theoretically, the possibility that the reinnervation of tissue biopsies (figure 1 form ref. 16) derives form fibers others than those degenerated because of capsaicin treatment (see Neuroscience. 1990;39(2):501-11. doi: 10.1016/0306-4522(90)90286-d).

REPLY:  We acknowledge that ablation, though used in the literature (indeed, used by several publications from the current authors), may not be an ideal term. We have adopted the term “disruption” and removed reference to ablation other than to indicate that our use of disruption is the same as ablation used by others.

As you have suggested, in many preclinical studies, it is well established that high doses of systemic capsaicin can induce neuronal death within sensory ganglia. However, peripheral or intrathecal administration of capsaicin at a lower doses produces only localized and temporary disruption (ablation) of axonal terminals without complete permanent afferent denervation of ganglia neurons. Since the therapeutic dose of capsaicin for treating pain in humans is far less than the amount required for the complete permanent denervation (neuronal death) in animals, it is highly unlikely that therapeutic administration of capsaicin induces permanent loss of sensory ganglia neurons. However, we agree that the regenerated fiber at the site of capsaicin injection can be derived from the disrupted axons or adjacent unaffected axons. We now acknowledge that reinnervation may include both the regeneration of affected fibers and ramification of adjacent unaffected fibers that are either TRPV1-positive or negative.

  1. When describing the results of IA capsaicin injection for treating osteoarthritis (page 5 line 10) the authors indicate as statistically significant a P=0.0740. This is not correct. Indeed, they correctly remark in the subsequent text that the results of 0.5 mg were simply indicative of an effect, but as such the text is confusing to readers.

REPLY: We have have not directly referred to 0.07 as being significant, but qualified the results by indicating that the alpha for this study was set at 0.1 a priori, so 0.07 was statistically significant as defined in the protocol. We hope this is acceptable.

  1. It should also be useful mentioning prospectively that preclinical studies have indicated the existence of a large proportion of polymodal C-fibers that do not express TRPV1 and it appears that different types of stimuli (thermal versus mechanical/chemical) can activate these TRPV1−negative non-peptidergic C-nociceptors (Cell Mol Neurobiol. 2020 Apr 18. doi: 10.1007/s10571-020-00847-w. Online ahead of print). If this may be translated clinically, it would imply that capsaicin will be ineffective in suppressing all types of pain when given in osteoarthritis or other conditions.

REPLY: This is a great point, thank you. We have added this information to the end of section 2.